# Polymer-Degrading Enzymes of *Pseudomonas chloroaphis* PA23 Display Broad Substrate Preferences

**DOI:** 10.3390/ijms24054501

**Published:** 2023-02-24

**Authors:** Nisha Mohanan, Michael C.-H. Wong, Nediljko Budisa, David B. Levin

**Affiliations:** 1Department of Biosystems Engineering, University of Manitoba, Winnipeg, MB R3T 5V6, Canada; 2Department of Chemistry, University of Manitoba, 144 Dysart Rd., Winnipeg, MB R3T 2N2, Canada; 3Biocatalysis Group, Technical University of Berlin, Müller-Breslau-Str. 10, D-10623 Berlin, Germany

**Keywords:** *Pseudomonas chlororaphis*, lipase, PHA depolymerase, polyhydroxyalkanoate, polylactic acid, poly(ε-caprolactone), polyethylenesuccinate, biodegradation

## Abstract

Although many bacterial lipases and PHA depolymerases have been identified, cloned, and characterized, there is very little information on the potential application of lipases and PHA depolymerases, especially intracellular enzymes, for the degradation of polyester polymers/plastics. We identified genes encoding an intracellular lipase (LIP3), an extracellular lipase (LIP4), and an intracellular PHA depolymerase (PhaZ) in the genome of the bacterium *Pseudomonas chlororaphis* PA23. We cloned these genes into *Escherichia coli* and then expressed, purified, and characterized the biochemistry and substrate preferences of the enzymes they encode. Our data suggest that the LIP3, LIP4, and PhaZ enzymes differ significantly in their biochemical and biophysical properties, structural-folding characteristics, and the absence or presence of a lid domain. Despite their different properties, the enzymes exhibited broad substrate specificity and were able to hydrolyze both short- and medium-chain length polyhydroxyalkanoates (PHAs), para-nitrophenyl (pNP) alkanoates, and polylactic acid (PLA). Gel Permeation Chromatography (GPC) analyses of the polymers treated with LIP3, LIP4, and PhaZ revealed significant degradation of both the biodegradable as well as the synthetic polymers poly(ε-caprolactone) (PCL) and polyethylene succinate (PES).

## 1. Introduction

The *Pseudomonadaceae* are a family of Gram-negative aerobic chemoheterotrophs that possess diverse enzyme systems and are capable of many biochemical transformations. Members of the metabolically diverse genus *Pseudomonas* have gained particular interest due to their capabilities to degrade and metabolize synthetic plastics [1]. *Pseudomonas chlororaphis* PA23 is a known biocontrol agent that exhibits plant growth promotion potential and inhibits the growth of plant fungal pathogens by secreting an arsenal of antibiotics and degradative enzymes [2,3].

Lipases (E.C.3.1.1.3) and Poly(3-hydroxyalkanoate) (PHA) depolymerases (EC 3.1.1.75 and EC 3.1.1.76) are carboxylic ester hydrolases belonging to the α/β-hydrolase fold family [4,5]. Lipases and PHA depolymerases differ greatly in sequence but have an analogous α/β-hydrolase fold and a catalytic triad consisting of serine/cysteine, histidine, and aspartate/glutamate.

Lipase enzymes are known to catalyze the hydrolysis of triacylglycerides (TAGs) to free fatty acids and glycerol. In addition, some lipases are able to synthesize new products by esterification, transesterification, and aminolysis [6,7]. Lipases act by “interfacial activation” where the aqueous phase containing the enzyme meets the insoluble, hydrophobic surface of the substrate [6]. This ability to function with enhanced activity at the nonpolar hydrophobic–aqueous hydrophilic interface is a property that separates lipases from other esterases [7].

These enzymes have been a kind of workhorse for decades to achieve polymer biodegradability, which would provide a solution to many global environmental issues and would also be useful for biomedical technologies. To advance the research, several aliphatic polyesters have been developed with properties comparable to those of conventional plastics. These materials serve as models for biodegradable plastics and are made from petrochemicals, such as poly(ethylene succinate (PES) or poly(ε-caprolactone) (PCL), or from biorenewable resources, such as short chain length- (scl-) or medium chain length- (mcl-) PHAs [8,9].

PHA depolymerases are commonly known to degrade polyhydroxyalkanoates (PHAs) [10]. PHA polymers synthesized by many microorganisms are a potential alternative to some petrochemical-based conventional plastics. PHAs are classified into two classes based on the number of carbon atoms in the monomers: scl-PHA polymers contain subunits with 6 to 5 carbon atoms per monomer, while subunits of mcl-PHA polymers contain 6 to 15 carbon atoms per monomer. Depending on the substrate and its physical state, PHA depolymerases are classified into four families: intracellular PHA depolymerases that degrade native intracellular PHA granules (nPHAscl depolymerases and nPHAmcl depolymerases) and extracellular PHA depolymerases that degrade denatured extracellular PHAs (dPHAscl depolymerases and dPHAmcl depolymerases).

Lipases and PHA depolymerases differ widely in molecular weight, substrate preferences, and catalytic activities. Lipase enzymes have numerous nonpolar residues that cluster around the active site of the protein structure [7]. Molecular modeling studies have shown that the arrangement of nonpolar amino acids at the active site facilitate the chemical interactions at the substrate cleavage site, resulting in enzyme catalysis [11,12]. Many lipases have a lid-like structure, which when closed sequesters the active site and prevents its interaction with the substrate. However, when the lid is open, the hydrophobic amino acid residues of the active site are exposed, resulting in hydrolysis of the polymer [7,13]. PHA depolymerases, esterases, and some lipases do not have lid-like structures [14,15].

Although many bacterial lipases and PHA depolymerases have been identified, cloned, and biochemically characterized [16,17], the ability of lipases and PHA depolymerases, especially intracellular enzymes, to degrade polyester polymers has not been well documented. Purified extracellular lipases from *Bacillus subtilis*, *Pseudomonas aeruginosa*, *Pseudomonas alcaligenes*, and *Burkholderiacepacia* (formerly known as *Pseudomonas cepacia*) have been shown to degrade polyesters polymers, such as PCL and poly-4-hydroxyalkaonaote (P(4HB)) [18]. PCL was also shown to be degraded by the extracellular lipase of a *Pseudomonas* sp. [19].Weak but detectable mcl-PHA activity was observed with the extracellular lipases from *P. alcaligenes* and *P. aeruginosa* [18]. The extracellular PHA depolymerases of *Streptomyces roseolus* SL3 and *Pseudomonas alcaligenes* LB19 hydrolyzed mcl-PHA and PCL. Hydrolysis ofmcl-PHA was also detected by the extracellular PHA depolymerases of *P. fluorescens* GK13 [20,21] and *Bdellovibriobacteriovorus* HD100 [22], as well as the intracellular PHA depolymerases of *Pseudomonas putida* KT2442 [18,23] and *Pseudomonas putida* LS46 [17].

In a previous report from our research group [24], it was shown that the cell-free culture supernatant of *Pseudomonaschlororaphis* PA23 can hydrolyze the ester bonds of p-nitrophenyl-alkanoate substrates with different carbon chain lengths as well as scl-PHA and mcl-PHA polymers with different subunit compositions. In the present study, two genes encoding an intracellular and an extracellular lipase and an intracellular PHA depolymerase gene were cloned from *P. chlororaphis* PA23 and expressed in *E. coli* BL21(DE3) to investigate the biochemical properties. We are particularly interested in evaluating their enzymatic activities in terms to their ability to hydrolyze various polyester polymers, such as scl- and mcl-PHAs, polylactic acid (PLA), poly(ethylene succinate) (PES), and PCL.

## 2. Results

### 2.1. Cloning, Expression and Purification of LIP3, LIP4 and PhaZ

The complete genome sequence of *Pseudomonas chlororaphis* PA23 is available in NCBI (https://www.ncbi.nlm.nih.gov (accessed on 1 July 2021)) and Integrated Microbial Genome (IMG) (https://img.jgi.doe.gov (accessed on 1 July 2021)) (Genome Acc. No. CP008696). Careful examination revealed seven genes encoding esterase/lipase enzymes in the *P. chlororaphis* PA23 genome (Table 1). Three of these encode intracellular lipases (EY04_08410, EY04_09635, and EY04_02420), while the other three contain a signal peptide indicative of possible extracellular activity (EY04_21540, EY04_17885, and EY04_32435). Curiously, only one encodes an intracellular PHA depolymerase (EY04_01535). In our initial experiments with *P. chlororaphis* PA23 and *Acinetobacter lwoffii*, we found that hydrolysis of the ester bonds of pNP-alkanoate substrates and the ester bonds of PHA polymers with different subunit compositions by the extracellular esterases/lipases was detectable in the growth medium [24]. In the present study, two genes encoding an intracellular and extracellular lipaseand an intracellular PHA depolymerase were cloned and expressed in *E. coli* BL21(DE3) to compare the properties and investigate their biodegradability compared to different biodegradable polymers.

We used the pET28a(+) vector to clone the *lip3*, *lip4*, and *phaZ* genes using *Escherichia coli* BL21(DE3) as the expression host. The main expression/fermentation parameter was that optimal expression was achieved after 5 h with 0.6 mmol L^−1^ IPTG at 37 °C (Figure 1A). After fermentation with expression, the cells were harvested and the recombinant proteins were purified to homogeneity. We roughly estimated the molecular masses by SDS-PAGE analysis of the purified enzymes and found molecular masses of 32 to 38 kDa for LIP3, LIP4, and PhaZ. These masses were consistent with the predicted molecular masses from their amino acid sequences (https://web.expasy.org/protparam (accessed on 15 September 2021)) (Figure 1A,B).

Purified enzymes were then applied to agar plates with colloidal overlays of different polymers to assess their polymer-degrading abilities. Control treatments consisted of soluble crude extracts of uninduced recombinant *E. coli* BL21 cells containing *lip3-pET28a(+)/lip4-pET28a(+)/phaZ-pET28a(+)*. Enzymatic hydrolysis of the polymers was verified by observing the formation of fluorescent “zones of clearance” around the wells after 48 h incubation at 30 °C under UV irradiation at 350 nm (Figure 2). These experiments clearly demonstrated that the purified enzymes were able to hydrolyze PHA polymers.

### 2.2. Sequence Analyses and Homology Modeling of LIP3, LIP4, and PhaZ

The *lip3*, *lip4*, and *phaZ* genes encode enzymes with predicted molecular masses of 32.5 kDa, 34.0 kDa, and 31.7 kDa, respectively. BLAST analyses of the deduced amino acid sequences (BLASTp) confirmed that the LIP3 and LIP4 enzymes belong to the lipase superfamily, whereas PhaZ belongs to the depolymerase superfamily, and all three have an α/β-hydrolase fold. Alignment of the amino acid sequences of LIP3 and LIP4 revealed very low amino acid sequence identity (12.29%), LIP3 and PhaZ had an amino acid sequence identity of 14.44%, and LIP4 and PhaZ showed an amino acid sequence identify of 9.86%. However, all three enzymes have conserved amino acid sequence motifs corresponding to the lipase box (GX_1_SX_2_G) characteristic of the lipase/esterase family and the putative oxyanion hole residues (GMLG- in LIP3; HGGGW- in LIP4, and PFV- in PhaZ) in the catalytic domain (Figure 3, Figure 4 and Figure 5). The x_1_ sites in the lipase boxes of LIP3, LIP4, and PhaZ are occupied by histidine (H82), asparagine (N179), and valine (V101), while the x_2_ sites contain glutamine (Q84), valine (V181), and tryptophan (W103), respectively. The enzymes possess a catalytic triad (serine-histidine-aspartic acid) characteristic of other lipases. The catalytic triad LIP3 has Ser^83^-His^263^-Asp^241^ (Figure 3), the catalytic triad of LIP4 has Ser^180^-His^307^-Asp^277^ (Figure 4), while PhaZ has Ser^102^-His^248^-Asp^221^ in its active site (Figure 5).

PSI-BLAST search for LIP3, LIP4, and PhaZ in the protein data bank revealed several bacterial lipases and PHA depolymerases as close homologues. Lipase from *Proteus mirabilis* (4GW3) was chosen as the template for LIP3 (43% amino acid identity and 95% similarity), lipase CinB from *Enterobacter asburiae*, which is an acetyl esterase (6KMO), was chosen for LIP4 (67% amino acid identity and 98% similarity), and PHA depolymerase from *Pseudomonas oleovorans* was chosen for PhaZ (91% amino acid identity and 98% similarity).

Appendix A show the homology model of LIP3, LIP4, and PhaZ, respectively, with the substrate 3-hydroxyoctanoate. The core domain of LIP3 consists of a five-parallel β-sheet, whereas LIP4 and PhaZ consists of an eight-parallel β-sheet (except for strand two, which is antiparallel).In each model, there are multiple α-helices on each side of the core β-sheet. Appendix A show the binding pocket of the substrate 3-hydroxyoctanoate dimer (residues 130–148 and 221–232 of LIP3, and 23–63 of LIP4), while the substrate binding pocket of PhaZ is exposed and lacks the closing lid domain.

To better illustrate the substrate binding pocket, the closing lid domain is not shown in Appendix A. It should also be noted that Appendix A present the surface of the lipases and PhaZ, clearly showing that the substrate binding pocket for LIP3 and LIP4 is buried by the lid domain. The binding energy for the substrate to LIP3, LIP4, and PhaZ determined in the docking experiment was calculated to be −6.2 kcal/mol, −5.8 kcal/mol, and −5.7 kcal/mol, respectively.A negative free energy of binding means that the substrates dock spontaneously.

### 2.3. Characterization of the Recombinant LIP3, LIP4, and PhaZ

LIP3, LIP4, and PhaZ were active over a wide pH range (pH 4.0 to 10.0) with optimal activity at pH 8.0 for LIP3 and PhaZ, and pH 6.0 for LIP4 (Figure 6A). LIP3 and PhaZ retained more than 50% of their activity over a pH range from 7.0 to 9.0, with PhaZ showing higher relative activity at pH 8.0–10.0. However, LIP4 retained more activity (>80%) atpH 5.0–6.0 compared with activity at an alkaline pH 8.0–10.0 (<40%).

LIP3, LIP4, and PhaZ were optimally active at 45 °C to 50 °C and retained more than 70% activity when the reaction was performed at 55 °C (Figure 6B). In addition, thermal inactivation experiments showed that the enzymes retained nearly 70% activity for 24 h at 45 °C (Figure 6C). LIP3 and PhaZ, on the other hand, were very stable and retained more than 80% of their activity for 24 h at 45 °C. LIP3 exhibited higher activation energy (E_a_ = 28.2 KJ mol^−1^) than LIP4 (26.4 KJ mol^−1^) and PhaZ (19.9 KJ mol^−1^) for the thermo-inactivation process within the temperature range of 25–50 °C.

When pNP-octanoate was used as a substrate, the purified enzymes were found to be very stable and retained almost 100% of their activities after four months of incubation at 4 °C and two months of incubation at 30 °C. The pNP-octanoate was also used as a substrate to determine the kinetics of each enzyme. The V_max_ and K_m_ for LIP3 were 625.0 μM mg^−1^ min^−1^ and 0.375 mmol L^−1^, respectively; the V_max_ and K_m_ of LIP4 were 555.55μM mg^−1^ min^−1^ and 0.333 mmol L^−1^, respectively; and the V_max_ and K_m_ of PhaZ were 909.09 μM mg^−1^ min^−1^ and 0.636 mmol L^−1^, respectively.

When the effect of cations and other compounds was examined (Table 2), it was found that 1 mmol L^−1^ CaCl_2_, 10 mmol L^−1^ EDTA and other metal ions did not affect the enzyme’s activity, indicating that the enzymes are not metalloenzymes and do not require metal ions (calcium, sodium, and magnesium) as cofactors for enzyme activity. PMSF and ionic and nonionic detergents (SDS and Tween 80) inhibited the activities of LIP3, LIP4, and PhaZ.

### 2.4. Substrate Specificity of LIP3, LIP4, and PhaZ

The lipase enzymes LIP3 and LIP4 as well as the PHA depolymerase PhaZ showed broad substrate specificity. In particular, the enzymes showed significant activity towards the ester bonds of pNP-alkanoates, which are the typical substrate for lipases (Figure 7). The enzymes hydrolyzed pNP-alkanoates with shorter and longer side chains, and the esterase activity for pNP octanoate was highest for LIP3 and PhaZ, whereas LIP4 showed the highest esterase activity for pNP-acetate. The substrate specificity of LIP3 and PhaZ for medium- and long-chain pNP-alkanoates was relatively higher than for short chain pNP-alkanoates. LIP4, on the other hand, showed higher specificity (>70%) for short chain pNP-alkanoates (pNP-acetate and pNP-butyrate).

Turbidimetric assays demonstrated that purified LIP3, LIP4, and PhaZ hydrolyzed ester linkages of β-polyhydroxyalkanoates/PHA polymers prepared from valeric acid (PHBV), hexanoic acid (PHHx), octanoic acid (PHO), nonanoic acid (PHN), and decanoic acid (PHD) (Table 3). The substrate preference for the LIP3 was PHHx > PHBV > PHO > PHN > PHD, the substrate preference for LIP4 was PHHx > PHO > PHN > PHD > PHBV, andthe substrate preference for PhaZ was PHO > PHN > PHD > PHHx > PHBV. Using PHO as substrate, LIP3 had a specific activity of 122.13 units*_PHA_* mg^−1^ protein, LIP4 had a specific activity of 102.6 units*_PHA_* mg^−1^ protein, and PhaZ had a specific activity of 650.13 units*_PHA_* mg^−1^ protein. The data reveal that the LIP3, LIP4, and PhaZ enzymes have a wide-range of substrate preferences, including scl-PHA polymers and mcl-PHA polymers, as well as C4 to C10 pNP-alkanoates. These enzymes also showed high activity towards PLA and were also able to hydrolyze synthetic polyester polymers, such as PCL and PES (Table 3).

### 2.5. Analysis of Degradation Products

We investigated the ability of the LIP3, LIP4, and PhaZ enzymes to degrade PHAs, PLA, and the petrochemical-based polymers, PCL and PES, using Gel Permeation Chromatography (GPC), which enabled quantification of changes in the polymer molecular weight (Mw), mass number (Mn), and molecular mass dispersivity (MP) after incubation with the enzymes. Encouragingly, the polymer samples before and after enzyme treatment showed significant differences. Figure 8, Figure 9 and Figure 10 show that for all GPC runs, no peak was observed after 11 min of elution. Overlaying the GPC chromatograms of PHBV, PHHx, PHO, PHN, PHD, PLA, PCL, and PES polymers after treatment with LIP3, LIP4, or PhaZ enzymes for 96 h, with chromatograms from untreated polymers, revealed significant changes in Mw, Mn, and MP values (see Figure 8, Figure 9 and Figure 10 and Appendix A).

The GPC data also support the observations that the LIP3, LIP4, and PhaZ enzymes were able to hydrolyze ester bonds of a wide variety of polymer types. Table 4–6 compares the percent change in molecular weights of the different polymers tested. These data indicate that molecular weight of PHN reduced by 49.7% after treatment by LIP3 and that LIP3 was able to reduce the molecular weights of PHHx, PHD, PLA, and PCL by similar percentages: 23.4%, 25.9%, 22.5, and 27.9%, respectively (Table 4). The molecular weight of PCL was reduced by 49.3% by LIP4, which was greater than the reduction in molecular weight of all of the PHA polymers tested and much greater than the reduction in molecular weight of PLA and PES (Table 5). PhaZ was also able to reduce the molecular weight of PCL by 27.6%, which was a far greater reduction in molecular weight caused by PhaZ for all the other polymers tested (Table 6).

## 3. Discussion

Analysis of the primary structures of the LIP3, LIP4, and PhaZ enzymes revealed no significant overall sequence homology between them. However, the lipase box consensus sequence motif GX_1_SX_2_G was found, with the X_1_ in LIP3 and LIP4 occupied by histidine and asparagine residues, respectively, while PhaZ had valine at the X_1_ position. In lipases and esterases, the X_1_ residue is usually occupied by a polar residue and X_2_ is variable [25]. In contrast, PHA depolymerases have a hydrophobic residue at position X_1_ [11,26]. Another conserved sequence region (GMLG- in LIP3, HGGGW- in LIP4 and PFV- in PhaZ) that appears to resemble the oxyanion hole consensus sequence in lipases and depolymerases was observed.

In general, lipases and PHA depolymerases are variable in size, with sequence similarity observable in short structural regions located around the active site residues [11,26]. In this context, Arpigny and Jaeger [27] suggested that bacterial lipases can be organized into eight classes based on their conserved amino acid sequence regions, structural-elements, and biochemical properties. The three-dimensional structures of lipases and PHA depolymerases show that these structures are very similar, despite sequence variability between them. Principally, they characteristically exhibit common motifs in their nuclei, known as α/β-hydrolase folds for the hydrolase and/or esterase families [28]. This common α/β-hydrolase fold consists of an eight-stranded, mostly parallel β-fold flanked by six α-helices, with a catalytic triad of amino acids (Ser/Asp-His-Asp/Glu). Interestingly, LIP3 has seven β-sheets with Ser83-His263-Asp241 in its active site, whereas LIP4 and PhaZ have eight β-sheets and a catalytic triad, Ser180-His307-Asp277 and Ser102-His248-Asp221, respectively.

The LIP3 and PhaZ enzymes were active in a pH range of 4.0 to 10.0 with optimal activity at pH 8 and 45–50 °C. This is consistent with the range reported for nPHAmcl depolymerase from *P. putida* KT2442 (pH 7.0 to 10.5) [29], as well as the lipase of *P. aeruginosa* MB 5001, which was reported to have a maximum esterase activity at pH 8.0 and 55 °C [30]. Similar to lipases from the psychrophilic *Pseudomonas fluorescens* strain AFT 36 [31] and *Pseudomonas* sp. with pH optima ranging from 6 to 9 [32] and lipases from *Pseudomonas* sp. and *P. fluorescens* AFT 36 with thermostabilities below 60 °C [31,33], LIP4 showed maximal activity at pH 6.0 and 50 °C.

Lipase PueB from *Pseudomonas chlororaphis* [34] and lipase from *Pseudomonas* sp. [35] showed stability at 100 °C. The thermal stability of LIP3 was relatively higher than that of LIP4 and PhaZ as concluded from the thermal inactivation studies. XtalPred analysis of the amino acid sequence revealed that the secondary structure of LIP3 had a slightly higher percentage of α-helix residues and random coils (88%) than that of LIP4 (82%) and PhaZ (84%). It is known that α-Helices confer conformational stability and random coils impart flexibility to the protein structure, which improves the thermal stability of the enzymes [36,37].

The activity of the LIP3, LIP4, and PhaZ enzymes were neither inhibited nor stimulated by the presence of EDTA and metal ions. These data suggest that metal ions are not required as cofactors for the activities of the three enzymes. The lipases of *Trichosporonasteroides* strain LP005 [38] and *Aspergillus terreus* [39] and the PHA depolymerase of *P. putida* LS46 [36,37] were also not influenced by the presence of metal ions.

The stimulatory effectof Ca^2+^ on activity was observed in lipases from *Pseudomonas putida* 3SK [40], *P. aeruginosa* EF2 [41], and *Pseudomonas* sp. 7323 [42]. It has been suggested that calcium ions are primarily involved in the removal of fatty acids formed as insoluble calcium soaps during hydrolysis, resulting in a change in the hydrophobic substrate–hydrophilic ratio in water [43]. It has also been suggested that the addition of calcium ions alters the orientation of the enzyme on the substrate molecule [44,45].

The inhibition of LIP3, LIP4, and PhaZ by phenylmethanesulphoylflouride (PMSF), a known inhibitor of enzyme activities, was comparable to that of PueB lipase from *Pseudomonas chlororaphis*, which showed 50% inhibition with 1 mmol L^−1^ PMSF [34]. The activities of LIP3, LIP4, and PhaZ were also inhibited by ionic and nonionic detergents, sodium dodecyl sulphate (SDS), and Tween 80indicating the presence of a hydrophobic region in the catalytic center and/or a change in the active configuration of the enzyme. Like PMSF, SDS is a known inhibitor of enzyme activities. It is reported to inhibit the lipase of *Pseudomonas putida* 3SK [40] and the PHA depolymerases of *P. fluorescens* GK13 [20] and *P. putida* LS46 [17].

It is known that many lipases exhibit a so-called interfacial activity observable at organic–aqueous interfaces and that many lipases have a mobile subdomain located, referred to as the “lid”, above the active site to facilitate this phenomenon [46,47]. The main function of the lid is to protect the active site, and it plays an important role in catalytic activity [48]. In aqueous media, the lid is closed, while in the presence of a hydrophobic layer, it allows access to the binding pocket. The lid of lipase has an amphipathic structure: in the closed state, its hydrophobic side faces the catalytic pocket, while the hydrophilic side faces the solvent [48]. The transition from the closed to the open conformation is accompanied by the exposure of the hydrophobic side to the substrate binding region [49]. Therefore, it is clear that the amphipathic nature of the lid and its specific amino acid sequence are important features for the enzyme activity and specificity of lipases [50]. However, PHA depolymerases and some lipases do not have a lid structure, such as the lipase from *Candida antarctica* B [15]. Modelling of the three-dimensional structures of LIP3, LIP4, and PhaZ complexed with a 3-hydroxyoctanoate dimer revealed that the core domains of LIP3 and LIP4 are of the α/β hydrolase-type and that their active sites are covered with a closing lid domain, whereas the PhaZdepolymerase lacks the lid domain.

The broad substrate preferences of LIP3, LIP4, and PhaZ have also been reported for the PueB lipase from *Pseudomonas chlororaphis*, which can also hydrolyze various pNP-alkanoates, such as pNP-acetate, pNP-propionate, and pNP-butyrate [46]. Indeed, LIP3 and PhaZ enzymes have the highest activity for pNP-octanoate, but different relative activities are observed for different polyesters. For example, LIP4 showed high relative activity towards pNP-acetate (C2) compared to LIP3 and PhaZ. On the other hand, PhaZ has a comparatively high substrate preference for mcl- PHA polymers. The differences in substrate specificity could be related to the different local microenvironment of the oxyanion pocket as well as to the configuration of the active site of the enzymes, which generate different affinities, activation barriers, and overall energetics of substrate binding.

The GPC data revealed degradation of a wide variety of polymer types by LIP3, LIP4, and PhaZ. A significant reduction in molecular weights of medium chain PHAs was observed with LIP3 (49.7%), while LIP4 and PhaZ exhibited significant degradation of PCL (49.3% and 27.6%) compared to other polymers, which clearly shows its high degradation potential compared to the known lipases/esterases. Kanmani et al. [51] reported a 21% molecular weight decrease of PHAs treated with *Bacillus subtilis* lipase, while PHA depolymerase from *P. putida* [17] and other lipases from *P. chlororaphis* [52] demonstrated 18–40% reduction in molecular weights of polymers after treatment with the enzymes.

PHAs are natural polymers synthesized by bacteria as carbon and energy storage compounds [53]. PHAs exists in two phases, amorphous and crystalline, similar to that observed for the bioceramics with highly variable (supervariate) compositions [54]. Intracellular native PHA polymer granules exhibit an amorphous state, while the PHA chains of extracted granules adopt crystalline phase [55].

The most prominent features of LIP3 and LIP4 that distinguishes them from other lipases described in the literature is their broad substrate specificity (Table 7). The enzymes LIP3, LIP4, and PhaZ were able to hydrolyze both PCL and PES to some extent. PCL is an aliphatic polyester consisting of repeating ω-hydroxyalkanoate units. Cell-free extracts of the yeast *Pseudozyma japonica* Y7-09 showed significant ability to degrade PCL [56]. Extracellular lipases of *P. aeruginosa* have shown the degradation of aromatic-aliphatic polyesters like PCL [1]. PES is a poly(alkene dicarboxylate) polymer structure, and while no studies in the published literature have reported degradation of PES by lipases [57], the intracellular, mcl-PHA depolymerase from *P. putida* LS46 was shown to have detectable activity towards PES [17]. 

## 4. Materials and Methods

### 4.1. Bacterial Strains and Plasmids

*P. chlororaphis* PA23 is a plant growth-promoting bacterium isolated from the rhizosphere of soybean that can suppress the growth of the fungal pathogen *Sclerotinia sclerotiorum* due to its ability to synthesize phenazine and pyrrolnitrine compounds [2,3]. *Escherichia coli* DH5a and *E. coli* BL21 (DE3) were used as host cells for recombinant plasmid construction and protein expression, respectively. Plasmid pET28a(+) (Novagen, Madison, WI, USA) was used as an expression vector for the expression of *lip3*, *lip4*, and *phaZ*. The PHA polymers used in the study were synthesized by *Pseudomonas putida* LS46 (International Depository Authority of Canada Accession Number 181110-03) [65,66].

### 4.2. Chemicals, Media and Growth Conditions

Hexanoic acid, octanoic acid, nonanoic acid, decanoic acid, PLA, PES, and PCL were from Sigma Chemical Co. (St. Louis, MO, USA). All other analytical grade products, chemicals, and dyes were from either Sigma Chemical Co. (St. Louis, MO, USA) or Fisher Scientific (Toronto, ON, USA). To prepare the various PHA polymers, the *P. putida* LS46 was cultivated in Ramsay’s minimal medium (RMM) [67] at 30 °C in a 7-LApplicon Bioreactor with hexanoic, octanoic, nonanoic, or decanoic acid as the sole carbon source [66]. A copolymer of 3-hydroxybutyrate and 3-hydroxyvalrate [Poly(3-hydroxybutyrate-co-3-hydroxyvalerate, PHBV] was prepared from biodiesel fatty acid supplemented with 0.5% valeric acid using *Cuprividusnecator* H16 [68]. The monomer compositions of these PHA polymers were determined [65] and are shown in Table 8. Luria Bertani (LB) broth and agar were used for the growth of *E. coli* DH5α and *E. coli* BL21 (DE3) cells.

### 4.3. DNA Manipulation and Plasmid Construction

The genomic DNA of *P. chlororaphis* PA23 was isolated from LB bacterial cultures using the Wizard DNA purification kit (Promega). DNA fragments containing genes encoding lipases (*lip3* and *lip4)* andPHA dpolymerase (*phaZ*) were generated separately by PCR amplification of *P. chlororaphis* PA23 chromosomal DNA using Phusion DNA polymerase (Thermo Scientific, Waltham, MA, USA),and the primer combinations are listed in Table 9. The nucleotide sequences of these genes were obtained from the whole genome sequence of *P. chlororaphis* PA23 from NCBI (Accession No. CP008696). Primers were designed to incorporate the restriction endonuclease sites NheI and HindIII at the 5′-ends of the forward and reverse primers, respectively. The signal peptide encoding region of *lip4* was not considered in the design of the primers.

PCR reactions were performed in a Thermocycler (BioRad, Hercules, CA, USA) under defined conditions (initial denaturation at 98 °C for 30 s followed by 34 cycles of 98 °C for 10 s, 66 °C for 30 s, and 72 °C for 1 min (min), with final extension of 10 min at 72 °C). The amplicon was cloned into the expression vector, pET-28a (+), and transformed into *E. coli* DH5α. The transformants were selected on LB kanamycin (50 µg/mL) plates. Subsequently, the recombinants were screened by colony PCR and restriction digestion to verify fall-out. The positive clones containing *lip3*, *lip4*, and *phaZ* were confirmed and checked for integration by sequencing at the Macrogen Nucleic Acid Sequencing Facility, Rockville, MD, USA.

### 4.4. Sequence Analysis and Homology Modeling

Amino acid sequence similarity searches for LIP3 (EY04_02420), LIP4 (EY04_21540), and PhaZ (EY04_01535) of *P. chlororaphis* PA23 were carried out using Basic Local Alignment Search Tool (BLAST) for proteins (http://blast.ncbi.nlm.nih.gov/Blast.cgi (accessed on 1 July 2021)). The sequences used were retrieved from the Integrated Microbial Genome (IMG) (https://img.jgi.doe.gov (accessed on 1 July 2021)) and/or NCBI (https://www.ncbi.nlm.nih.gov (accessed on 1 July 2021)). Alignment of amino acid sequences was performed using Clustal Omega (https://www.ebi.ac.uk/Tools/msa/clustalo/ (accessed on 1 July 2021)) and Espript 3.0 (http://espript.ibcp.fr/ESPript/ESPript/ (accessed on 1 July 2021)). Amino acid content was analyzed using the ProtParam tool (http://web.expasy.org/protparam/ (accessed on 15 September 2021)). Secondary structure predictions and other protein features were generated using the XtalPred server (http://ffas.burnham.org/XtalPred-cgi/xtal (accessed on 15 September 2021)). The three-dimensional structures of LIP3, LIP4, and PhaZ were modeled using SWISS-MODEL [69]. The lipase from *Proteus mirabilis* (4GW3) [70] was as a template for modelling LIP3. The acetyl esterase, CinB, from *Enterobacter asburiae* (6KMO) [71,72] served as a template for modelling LIP4, and the poly(3-hydroxyalkanoate) depolymerase from *Pseudomonas oleovorans* [73,74] (Alpha-Fold code AF-P26495-F1) was used as a template for modelling PhaZ. A dimer of 3-hydroxyoctanoic acid (HO) was generated using Avogadro, an open-source molecular builder and visualization tool, version 1.2.0 (http://avogadro.cc/ (accessed on 10 January 2022)) [75], and the universal force field [76] was used for energy minimization. The dimer was docked to the LIP3, LIP4, and PhaZ models using the Vina wizard in the PyRx software [77]. The figures were created using PyMOL (Molecular Graphics System, Version 2.5.2).

### 4.5. Expression and Purification of Recombinant LIP3, LIP4 and PhaZ

Recombinant plasmids carrying the cloned genes (*pET28a-lip3*, *pET28a-lip4*, and *pET28a-phaZ*) were isolated using the Geneaid plasmid isolation kit and introduced into *E. coli* BL21 (DE3) expression host. *E. coli* BL21 (DE3) cells harboring the recombinant plasmid were cultivated by inoculating 1% primary culture of the recombinant cells into 50 mL of LB broth supplemented with 50 µg/mL kanamycin. Host cells were grown in a shaking incubator at 37 °C until optical density of 0.6 at 600 nm (OD_600_) and induced with 0.6 mmol L^−1^ isopropyl β-D-1-thiogalactopyranoside (IPTG) for 5 h. Subsequently, the expression of *lip3*, *lip4*, and *phaZ* was examined at different IPTG concentrations (0.4, 0.6, 0.8, and 1.0 mmol L^−1^), growth temperatures (37 °C, 30 °C, and 16 °C) after induction, and incubation times (4, 8, and 16 h). E. coli BL21 (DE3) cells were harvested by centrifugation and lysed in chilled lysis buffer (25 mmol L^−1^ Tris-HCl, 10 mmol L^−1^ MgCl_2_, 100 mmol L^−1^ NaCl, and 1 mg/mL lysozyme) in an ultrasound machine (10 cycles with 1 min pulse (10 s on/off)). Recombinant enzymes were purified from the clear lysate (obtained after centrifugation at 10,000× *g* for 30 min) by affinity chromatography using HiTrap Ni^2+^-NTA resins (Qiagen) under nondenaturing condition according to the manufacturer’s instructions. Expression profiles and purity of recombinant proteins were analyzed by sodium dodecyl sulfate–polyacrylamide gel electrophoresis (SDS-PAGE).

### 4.6. Qualitative and Quantitative Estimation of Depolymerase/Esterase Activity of LIP3, LIP4, and PhaZ

To determine the depolymerase activity of the recombinant enzymes, a homogeneous latex suspension of PHA polymers synthesized by *P. putida* LS46 from hexanoic (PHHx), octanoic (PHO), nonanoic (PNO), and decanoic acids (PHD) was prepared as described by [29]. Four volumes of a PHA solution in acetone (0.1%, *w*/*v*) were added dropwise to one volume of cold water (5–10 °C) with stirring. The acetone was removed with speed vacuum concentrators to obtain a white colloidal suspension. PHA agar plates were prepared using 10 mg PHA polymer suspension and 1.5% (wt/vol) agar in 50 mmol L^−1^ phosphate buffer (pH 7). The enzyme solution was dropped into 5-mm-diameter wells punched into the PHA agar plates and then incubated at 30 °C for 48 h. The free (clear) zone around the wells indicated depolymerase activity. The fluorescent dye Rhodamine B (0.0005%) was added to the plates to show fluorescent halos around the wells when irradiated with UV light (λ = 350 nm) [77].

The PHA depolymerase activity of the enzyme was determined by measuring the decrease in turbidity of the PHA suspension at OD_650_ [62]. The reaction mixture contained 1 mg PHA latex, 50 mmol L^−1^ phosphate buffer, pH 7, and 0.025 mg protein solution. The turbidity decrease was measured every 24 h for 96 h. One unit of depolymerase activity by turbidimetric assay was determined as the amount of enzyme that can decrease the absorbance (OD_650_) by 1 absorbance unit per min (i.e.,1OD_650_/min). Alternatively, one unit of depolymerase activity (units*_PHA_*) was the amount of enzyme that hydrolyzed 1 μg of PHA in 1 min. Parallel PHA control samples (without the enzyme) were taken. All experiments were performed in triplicate.

The esterase activity of the enzymes was determined as described by Schirmer et al. [20] using p-nitrophenyl (pNP)-alkanoates with different carbon chain lengths as substrate. The recombinant enzymes from crude cell lysates and/or the purified enzymes (0.025 mg) were used in reactions with 3.9 mL PBS buffer and 100 µL substrate, p-NP-alkanoates (1 mmol L^−1^) at optimum temperature for 20 min. The reaction was stopped with 1 mL of 1 M sodium carbonate and the absorbance was recorded at OD_420_. One unit of esterase activity was measured as the amount of enzyme releasing 1 µmol of p-nitrophenol per min under optimal conditions. Total protein was determined by the Bradford assay, using Bovine Serum Albumin (BSA) as the standard protein [78].

### 4.7. Biophysical and Biochemical Properties and Substrate Specificity of LIP3, LIP4 and PhaZ

To determine the effect of pH on the activity of LIP3, LIP4, and PhaZ, the substratep-NP-octanoate (PNPO, 1 mmol L^−1^) was dissolved separately in buffers with different pH values [sodium acetate buffer (50 mmol L^−1^, pH 4.0–5.0), sodium phosphate buffer (50 mmol L^−1^, pH 6.0–8.0) and glycine-NaOH buffer (50 mmol L^−1^, pH 9.0–10.0)] and then used in reaction mixtures with the appropriately diluted enzyme (0.025 mg) solutions prepared in the desired buffers. The effects of temperature on enzyme activity were studied by testing at different temperatures (25–55 °C) for 20 min. The enzymes were exposed to different temperatures (30–60 °C) for a period of 24 h (h) to check their thermal stability. The activation energy of thermal inactivation for the enzymes was calculated from the slope (−E_a_/R) of Arrhenius plots (k_inact_ = Ae^−Ea/RT^), where A represents the Arrhenius constant, E_a_ is the activation energy, R is the gas constant, and T is the absolute temperature.

The effect of cations (chloride/sulfate salts), chelators (EDTA), and ionic and non-ionic detergents on the enzyme activities of LIP3, LIP4, and PhaZ was investigated by adding them to the reaction mixtures and incubating them in 50 mmol L^−1^ sodium phosphate buffer at the optimal pH and temperature conditions for 20 min. The residual enzyme activities were determined.

Substrate specificity of the enzymes was estimated for PHAs, PHBV, PLA, p-NP- alkanoate substrates (pNP-acetate, pNP-butyrate, pNP-octanoate, and pNP-decanoate), and petrochemical based polyester polymers, such as PES and PCL, at a substrate concentration of 1 mg under optimal conditions for enzyme activity. The V_max_ and K_m_ values were calculated using the Lineweaver–Burk linear regression plot. The purified enzymes, LIP3, LIP4, and PhaZ were incubated with pNP-octanoate substrate in concentrations ranging from 0.1 to 3.0 mmol L^−1^ under optimal assay conditions.

### 4.8. Gel Permeation Chromatography

The molecular weights of PHAs before and after degradation with LIP3, LIP4, and PhaZ were analyzed by gel permeation chromatography (GPC). The reaction mixture with different 10 mg polymer suspension (PHB, PHHx, PHO, PHN, PHD, PLA, PES, and PCL), prepared according to Schirmer et al. [20], was enzymatically hydrolyzed with the recombinant enzymes (2.5 mg) in separate experiments. The polymer substrates (without the enzyme) served as controls. After 96 h of incubation at optimal temperature, the reaction mixtures and substrate controls were oven dried at 60 °C. After cooling, samples were dissolved in chloroform to achieve a final sample concentration of 1.5 mg/mL and filtered with 0.45 mm PTFE. All test and control sample sets were analyzed by gel permeation chromatography (GPC) using a Waters Model 1515 solvent pump with a Waters Refractive Index detector (model 2414) and the Agilent PLgel MIXED-C column (7.5 mm i.d.: 1.5 mL/min). Data were acquired and analyzed using the Breeze 2 software, version (Waters Chromatography, Kent, UK). The procedure was calibrated with Agilent Polystyrene EasiCal PS-1 standards (Agilent Technologies, Mississauga, ON, Canada). The mobile phase (HPLC grade chloroform) was run at a flow rate of 1 mL/min. The column temperature was set at 30 °C and the sample injection volume was set at 20 µL. The method was integrated with the standards to obtain peak-average molecular weight (MP), number-average molecular weight (Mn), weight-average molecular weight (Mw), and polydispersity index (Mw/Mn).

## 5. Conclusions

In the present study, the remarkable versatility of the lipases and PHA depolymerase in terms of their potential application to the degradation of various polyesters, including pNP-alkanoates, PHAs, PLA, and to some extent the petrochemical-based polyester polymers, PCL and PES, have been demonstrated. Our data suggest that LIP3, LIP4, and PhaZ enzymes differ significantly in their biochemical and biophysical properties, structural folding with the substrate, and absence or presence of the lid domain.

## Figures and Tables

**Figure 1 ijms-24-04501-f001:**
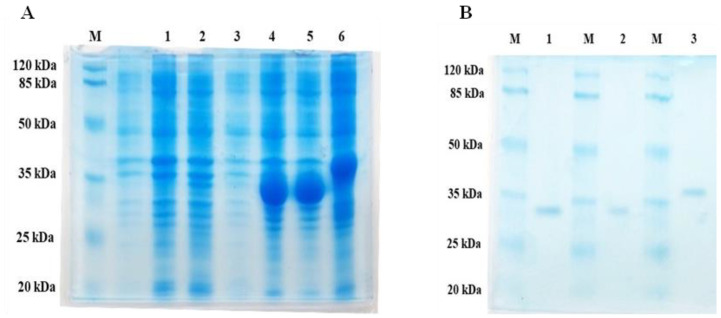
Expression and purification profiles (12% SDS-PAGE) of recombinant LIP3, LIP4, and PhaZ. (**A**) Cell lysates of *Escherichia coli* BL21 (DE3) transformed with *pET28a-phaZ/lip3/lip4* Lane 1–3, cell lysate of uninduced *E. coli* BL21 (DE3) carrying *phaZ-pET28a*, *lip3-pET28a* and *lip4-pET28a*; Lane 4, 5, 6, the lysate of the induced host cells containing *phaZ-pET28a*, *lip3-pET28a* and *lip4-pET28a* expressing PhaZ, LIP3, and LIP4, respectively.(**B**) SDS-PAGE-purification profile after His–Tag affinity chromatography: PhaZ (Lane 1), LIP3 (Lane 2), and LIP4 (Lane 3); Lane M, standard molecular weight marker (Thermo Fisher Scientific, Waltham, MA, USA).

**Figure 2 ijms-24-04501-f002:**
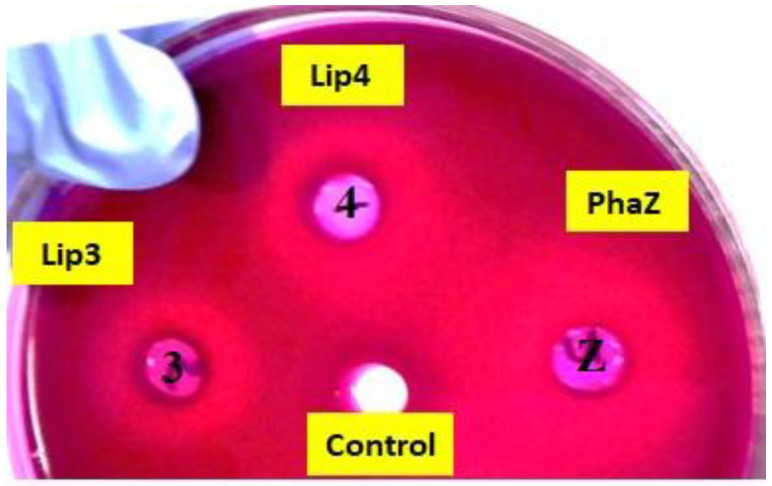
PHA agar plate assay for testing the depolymerase/esterase activity of LIP3, LIP4, and PhaZ. The clearance zones around the wells to which 20 µg of purified LIP3, LIP4, and PhaZ enzymes were added show hydrolysis of mcl-PHA in rhodamine B-PHO agar plates after UV irradiation. As a control, a crude extract of the noninduced expression host *E. coli* BL21(DE3) was used.

**Figure 3 ijms-24-04501-f003:**
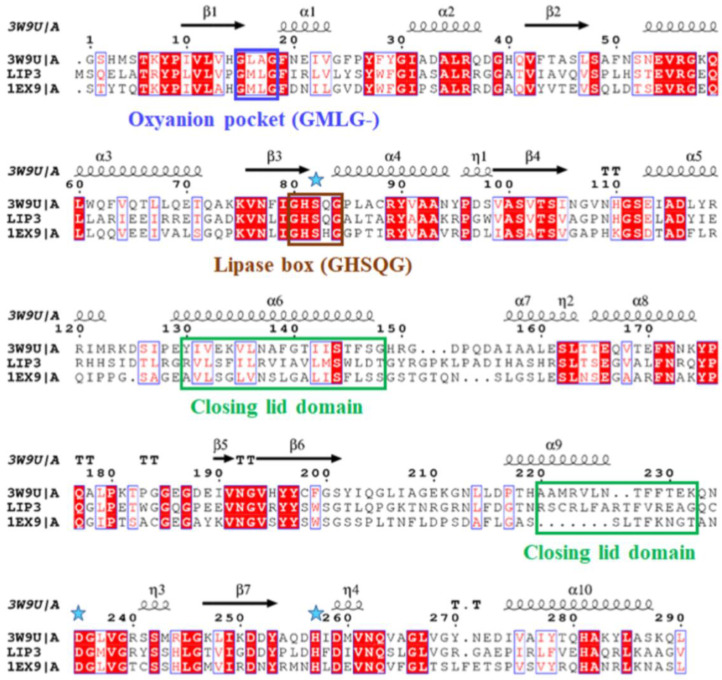
Amino acid sequence alignment of LIP3 with lipases from *Proteus mirabilis* HI4320 (3W9U) and *Pseudomonas aeruginosa* (1EX9). Brown box: amino acids of the lipase box consensus sequence; Blue box: amino acid residues of the oxyanion hole; green box: amino acid residues of the closing lid domain; blue stars: show the positions of the amino acids of the catalytic triad (serine, glutamic acid, and histidine); secondary structure α-helix and β-sheet regions are shown alone at the top of the alignment. White letters with red highlights represent amino acid residues that are conserved in all the enzyme sequences and red letters in blue frame indicate conserved regions with identical amino acid residues (residues with similar properties) in one or two of the enzyme sequences.

**Figure 4 ijms-24-04501-f004:**
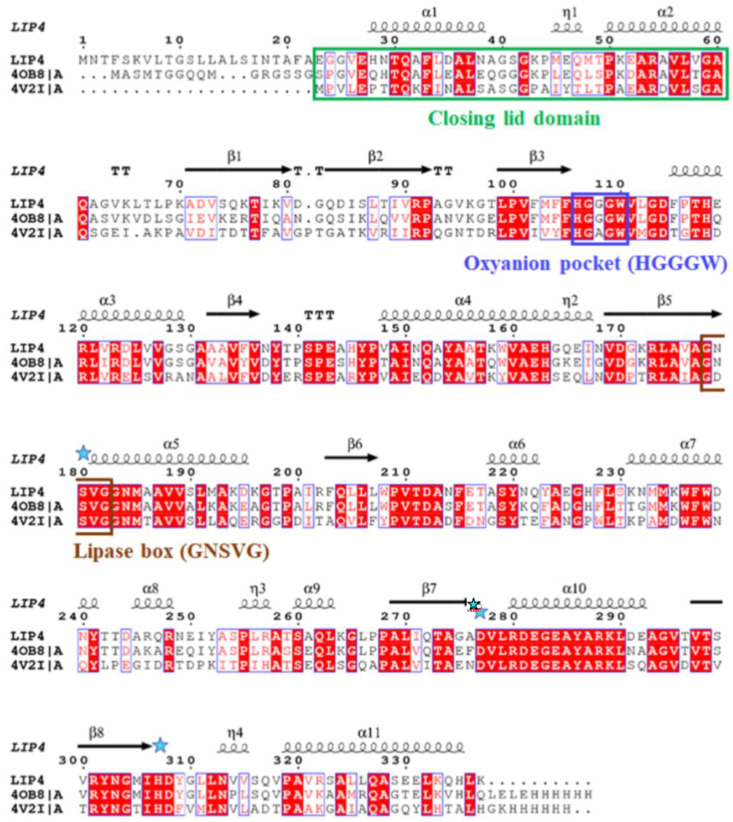
Amino acid sequence alignment of LIP4 with the alpha/beta hydrolase fold-3 domain protein from *Pseudomonas* sp. ECU1011 (4OB8) and lipase of *Thalassospira* sp. GB04J01 (4V2I). Brown box: amino acids of the consensus sequence of the lipase box; blue box: amino acid residues of the oxyanion hole; green box: amino acid residues of the closing lid domain; blue asterisks: the positions of the amino acids of the catalytic triad (serine, glutamic acid, and histidine) in the sequence; the secondary structure segments of the α-helix and the β-sheet are shown alone at the top of the alignment. White letters with red highlights represent amino acid residues that are conserved in all the enzyme sequences and red letters in blue frame indicate conserved regions with identical amino acid residues (residues with similar properties) in one or two of the enzyme sequences.

**Figure 5 ijms-24-04501-f005:**
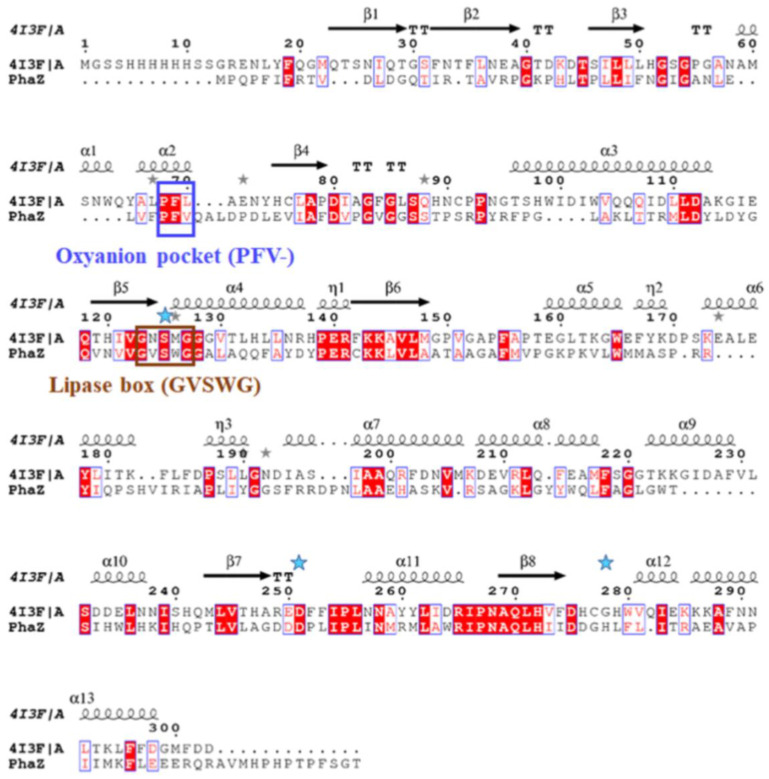
Amino acid sequence alignment of PhaZ with the serine hydrolase CCSP0084 from *Cycloclasticus* sp. P1 (4I3F). Brown box: amino acids of the Lipase box consensus sequence; blue box: amino acid residues of the oxyanion hole; blue asterisks indicate the positions of the amino acids of the catalytic triad (serine, glutamic acid, and histidine); the sequence parts corresponding with α-helix and β-sheet regions are shown alone at the top of the alignment. White letters with red highlights represent amino acid residues that are conserved in all the enzyme sequences and red letters in blue frame indicate conserved regions with identical amino acid residues (residues with similar properties) in one or two of the enzyme sequences. The grey asterisks above amino acid residues 67, 88, 126, and 192 of both sequences, 75 and 126 of the PhaZ sequence, 174 of the 4I3F/A sequence, indicate positions with amino acids with similar properties.

**Figure 6 ijms-24-04501-f006:**
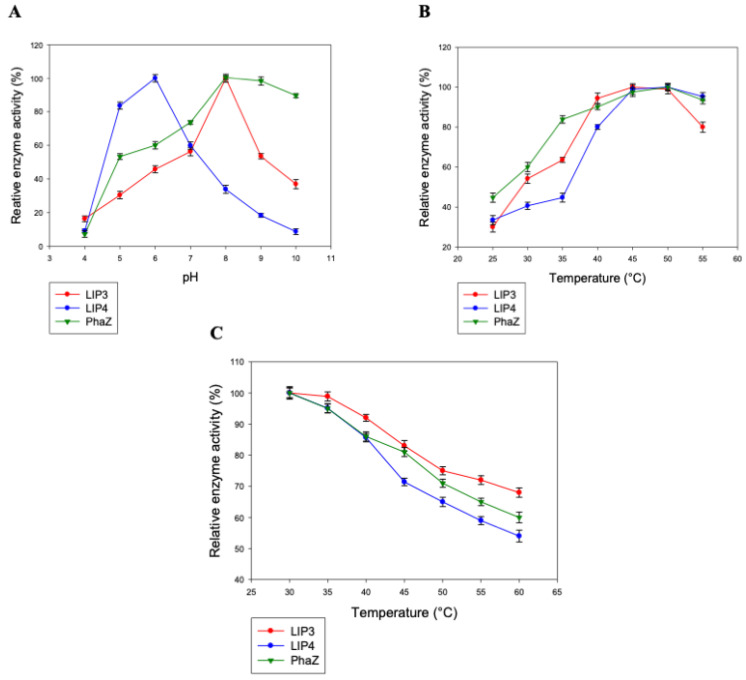
Determination of the parameters of enzymatic activity of LIP3, LIP4, and PhaZ. The effects of (**A**) pH and (**B**) temperature on the activity of LIP3 (red line), LIP4 (blue line), and PhaZ (green line). Recombinant LIP3 was optimally active at pH 8.0 and 45 °C, LIP4 at pH 6.0 and 50 °C, and PhaZ showed maximum activity at pH 8.0 and 50 °C. The observed maximum activity was taken as 100%. (**C**) The red line shows the effect of temperature on the stability of LIP3; the blue line the effect of temperature on the stability of LIP4; the green line shows the effect of temperature on the stability of PhaZ. Thermal stability of enzymes was measured over a period of 24 h (h) at various temperatures (30–60 °C). As a control, the activity of the untreated enzyme (0.025 mg/mL) was set at 100%.

**Figure 7 ijms-24-04501-f007:**
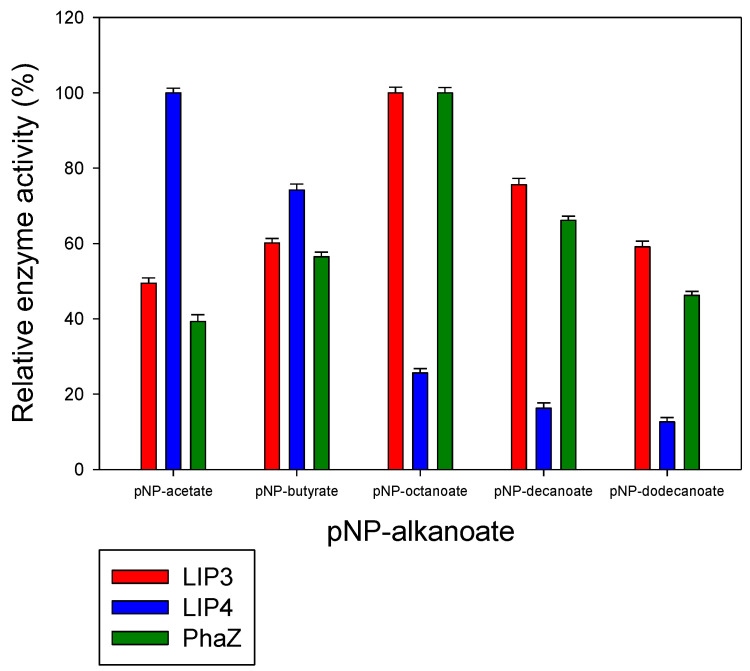
Determination of the activities of LIP3, LIP4, and PhaZ towards different substrates. Substrate specificity of lipases LIP3 (red bars), LIP4 (blue bars), and PhaZ (green bars) from *P. chlororaphis* PA23. The reaction mixture contained 1 mmol L^−1^ of various p-nitrophenyl (pNP) alkanoates and 0.025 mg of purified enzyme under optimal conditions of the enzymes. The maximum activity observed was taken as 100%. The 100% activity for LIP3, LIP4, and PhaZ corresponded to 561.87 U mg^−1^, 453.72 U mg^−1^, and 860.13 U mg^−1^ for pNP-octanoate, respectively.

**Figure 8 ijms-24-04501-f008:**
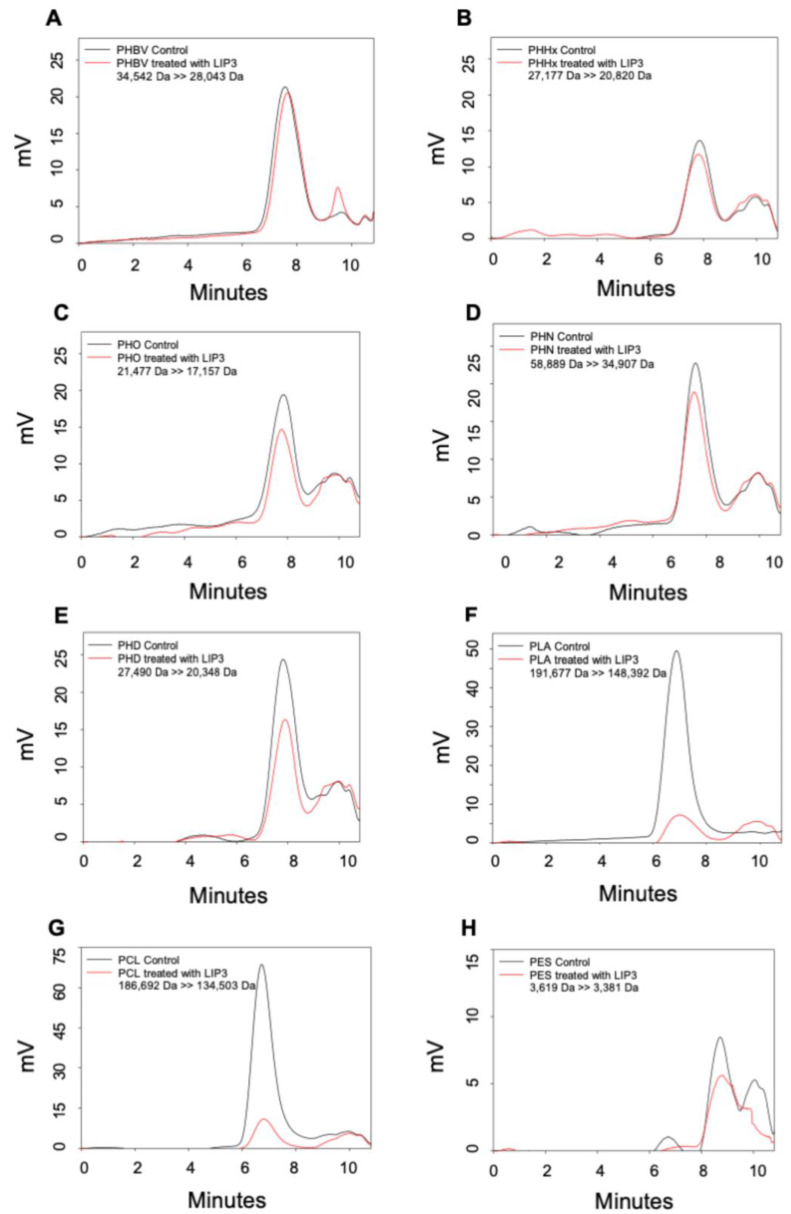
Gel Permeation Chromatography (GPC) analyses of LIP3. The overlay of chromatograms of different polymer substrates before (control = black line) and after LIP3 treatment (red line). (**A**) PHBV; (**B**) PHHx; (**C**) PHO; (**D**) PHN; (**E**) PHD; (**F**) PLA; (**G**) PCL; and (**H**) PES. Numbers indicate the change in polymer molecular weight before and after treatment. mV, the HPLC Refractive Index Detector records the signa intensity in millivolts.

**Figure 9 ijms-24-04501-f009:**
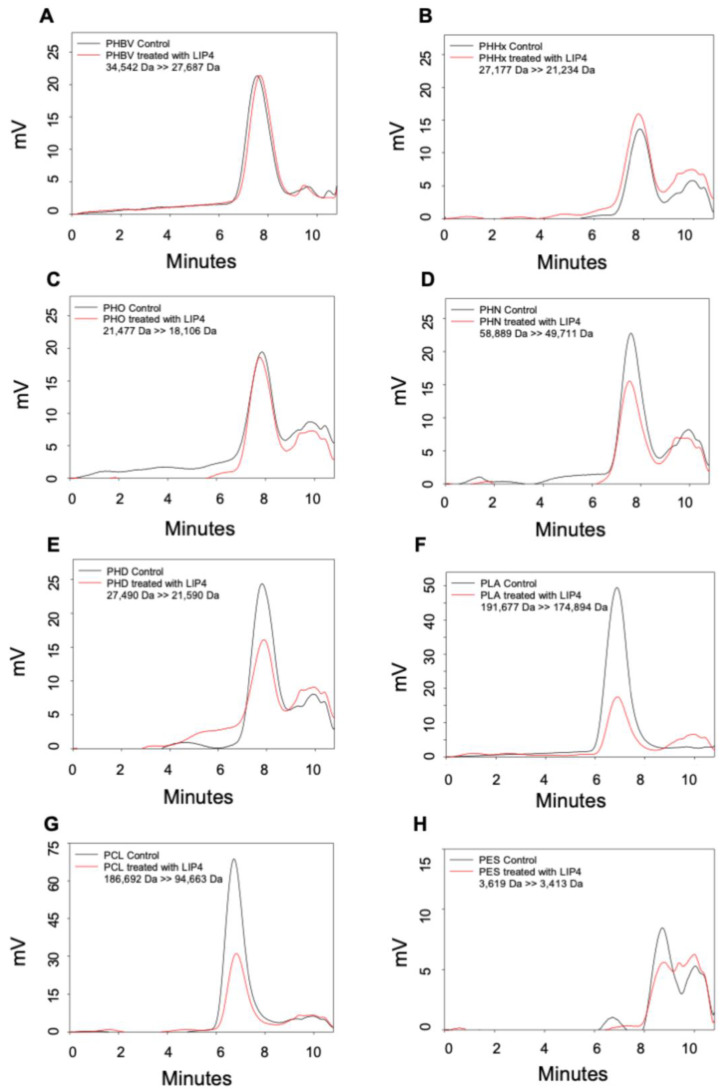
Gel Permeation Chromatography (GPC) analyses of LIP4. The overlay of chromatograms of different polymer substrates before (control = black line) and after LIP4 treatment (red line). (**A**) PHBV; (**B**) PHHx; (**C**) PHO; (**D**) PHN; (**E**) PHD; (**F**) PLA; (**G**) PCL; and (**H**) PES. Numbers indicate the change in polymer molecular weight before and after treatment. mV, the HPLC Refractive Index Detector records the signa intensity in millivolts.

**Figure 10 ijms-24-04501-f010:**
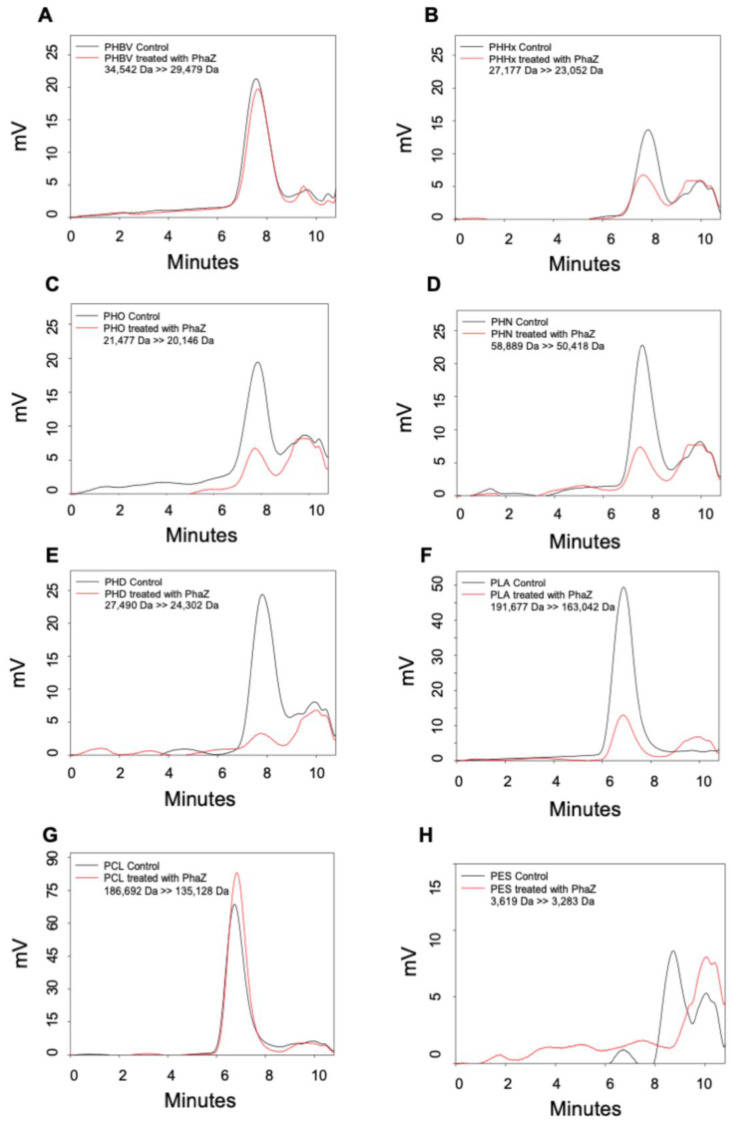
Gel Permeation Chromatography (GPC) analyses of PhaZ. The overlay of chromatograms of different polymer substrates before (control = black line) and after PhaZ treatment (red line). (**A**) PHBV; (**B**) PHHx; (**C**) PHO; (**D**) PHN; (**E**) PHD; (**F**) PLA; (**G**) PCL; and (**H**) PES. Numbers indicate the change in polymer molecular weight before and after treatment. mV, the HPLC Refractive Index Detector records the signa intensity in millivolts.

**Table 1 ijms-24-04501-t001:** Putative lipases and PhaZ present in the genome of *Pseudomonas chlororaphis* PA23.

Lipases	Accession Number	Location	Gene Size (bp)	Number of Amino Acids (aa)	Molecular Weight (kDa)
LIP1	EY04_08410	Intracellular	828	275	29.7
LIP2	EY04_09635	Intracellular	834	277	30.2
LIP3	EY04_02420	Intracellular	891	296	32.5
LIP4	EY04_21540	Extracellular	942	315	34.0
LIP5	EY04_17885	Extracellular	1905	634	69.8
LIP6	EY04_32435	Extracellular	1911	636	70
PhaZ	EY04_01535	Intracellular	858	285	31.7

**Table 2 ijms-24-04501-t002:** Effects of different modulators on the activity of LIP3, LIP4, and PhaZ.

Modulators/Reagents	Final Concentration	Relative Activity (%) *		
		LIP3	LIP4	PhaZ
EDTA	10 mmol L^−1^	94 ± 1.0	98 ± 1.8	97 ± 0.9
CaCl_2_	1 mmol L^−1^	93 ± 2.5	91 ± 3.3	94 ± 1.6
MgCl_2_	1 mmol L^−1^	96 ± 2.1	94 ± 2.6	74 ± 2.7
NaCl	50 mmol L^−1^	90 ± 3.8	89 ± 3.1	78 ± 3.6
PMSF	1 mmol L^−1^	71 ± 4.5	67 ± 4.5	59 ± 3.1
Tween80	0.1%	47 ± 3.8	36 ± 3.5	53 ± 2.3
SDS	5%	30 ± 0.6	34 ± 1.5	28 ± 1.1
Enzyme activity (without modulator)		100	100	100

* The values shown are the averages of three independent (biological) replicate experiments ± standard deviations.

**Table 3 ijms-24-04501-t003:** Substrate specificity of purified LIP3, LIP4, and PhaZ.

Substrate	Relative Activity (%) *		
	LIP3	LIP4	PhaZ
PHA polymers			
PHBV	70.82 ± 3.6	48.78 ± 2.9	43.64 ± 5.1
PHHx	100	100	76.70 ± 3.6
PHO	57.40 ± 1.9	86.25 ± 5.2	100
PHN	54.52 ± 4.6	74.92 ± 1.6	91.86 ± 4.2
PHD	52.07 ± 5.2	61.44 ± 4.7	88.23 ± 2.9
Other biodegradable/petrochemical-based polymers			
Polylactic acid (PLA)	53.5 ± 2.7	33.16 ± 2.1	40.42 ± 3.7
Poly(ε-caprolactone) (PCL)	25.43 ± 3.9	47.65 ± 2.6	38.90 ± 4.1
Poly(ethylene succinate) (PES)	13.21 ± 4.1	11.32 ± 4.1	22.76 ± 4.1

* Values shown are the averages of three independent (biological) replicate experiments ± standard deviations. The pure enzyme (0.025 mg) was tested with the indicated substrates at a final concentration of 1 mg for 96 h in all cases. The polymer without enzyme was used as a control. The maximum observed activity was set at 100%. Activity of 100% corresponded to 207.41 U mg^−1^ in LIP3, 114.09 U mg^−1^ in LIP4, and 650.13 U mg^−1^ in PhaZ.

**Table 4 ijms-24-04501-t004:** Percent decrease in polymer molecular weight after treatment with LIP3.

Polymer Substrate	Mw before (Da)	Mw after (Da)	Mw (Da)	% Change
PHBV	34,542	28,043	6499	18.8
PHHHx	27,177	20,820	6357	23.4
PHO	21,477	17,157	4320	20.1
PHN	58,889	34,907	23,982	49.7
PHD	27,490	20,348	7142	25.9
PLA	191,677	148,292	43,285	22.5
PCL	186,692	134,503	52,189	27.9
PES	3619	3381	238	6.6

**Table 5 ijms-24-04501-t005:** Percent decrease in polymer molecular weight after treatment with LIP4.

Polymer Substrate	Mw before (Da)	Mw after (Da)	Mw (Da)	% Change
PHBV	34,542	27,687	6855	19.8
PHHHx	27,177	21,234	5943	21.9
PHO	21,477	18,106	3371	15.7
PHN	58,889	49,711	9178	15.6
PHD	27,490	21,590	5900	21.5
PLA	191,677	174,894	16,783	8.7
PCL	186,692	94,663	92,092	49.3
PES	3619	3414	205	5.7

**Table 6 ijms-24-04501-t006:** Percent decrease in polymer molecular weight after treatment with PhaZ.

Polymer Substrate	Mw before (Da)	Mw after (Da)	Mw (Da)	% Change
PHBV	34,542	29,479	5063	14.7
PHHHx	27,177	23,052	4125	15.2
PHO	21,477	20,146	1331	6.2
PHN	58,889	50,418	8471	14.4
PHD	27,490	24,302	3188	11.6
PLA	191,677	163,042	28,635	14.9
PCL	186,692	135,128	51,564	27.6
PES	3619	3283	336	9.3

**Table 7 ijms-24-04501-t007:** Degradation of polyesters by bacterial lipases and PHA depolymerases.

Bacterial Source	Lipases/ Esterases	Polyesters Hydrolyzed/Degradation Ability of the Enzymes	Polyesters NotHydrolyzed	Method Used	Activity	Ref.
*Bacillus subtilis*, *Pseudomonas aeruginosa*, *Pseudomonas alcaligenes*, *Burkholderiacepacia* (former *Pseudomonas cepacia*)	Extracellular lipases	pNP-palmitate Poly(6-hydroxyhexanoate) Poly(4-hydroxybutyrate) Polycaprolactone	Poly(3-hydroxybutyrate) Poly(3-hydroxyalkanoates) Polylactic acid	Turbidimetric assay; Rhodamine agar plate assay.	*B. subtilis* lipase: 0.2 × 10^3^ U/mg (pNPP) 6000 U mg^−1^ (PCL) *P. aeruginosa* lipase: 52 × 10^3^ U mg^−1^ (pNPP) 1.8 × 10^6^ U mg^−1^ (PCL) *P. alcaligenes* lipase: 8 × 10^3^ U mg^−1^ (pNPP) 140,000 U mg^−1^ (PCL) *B. cepacia* lipase: 0.5 × 10^3^ U mg^−1^ (pNPP) 40,000 U mg^−1^ (PCL)	[18]
*P. fluorescens* GK13	Extracellular esterase	vvpNPP	Polyhydroxyalkanoates Polycaprolactone Polylactic acid	Turbidimetric assay; Rhodamine agar plate assay.	0.4 × 10^3^ U mg^−1^ (pNPP)	[18]
*Bacillus subtilis*	Extracellular lipase	pNPP Polyhydroxyalkanoates	-	Molecular weight decrease by GPC; FTIR; NMR; DSC.	21.3% molecular weight decrease 28.3% weight loss 273.65 U mg^−1^ (pNPP)	[51]
*Geobacilluszalihae*	Extracellular lipase	Poly(3-hydroxybutyrate)	-	Poly(3-hydroxybutyrate) agar plate;	Clear zone around the colony	[58]
*Pseudomonas chlororaphis* PA23-63-1	Extracellular lipases and esterases mix	pNP-alkanoate Poly (3-hydroxybutyrate-co-3-hydroxyvalerate) Poly(3-hydroxyhexanoate) Poly(3-hydroxyoctanoate) Poly(3-hydroxynonanoate) Poly(3-hydroxydecanoate)	Polycaprolactone Polyethylene sulfonate	PHA agar plate assay; Turbidimetric assay; Weight loss of PHA films.	4.5% weight loss 997.7 U mg^−1^ (pNPO) 722 U mg^−1^ (PHO) Clear zone of PHA hydrolysis	[24]
*Pseudomonas chlororaphis* PA23	Intracellular lipases, LIP1, and LIP2	Poly (3-hydroxybutyrate-co-3-hydroxyvalerate Poly(3-hydroxyhexanoate) Poly(3-hydroxyoctanoate) Poly(3-hydroxynonanoate) Poly(3-hydroxydecanoate) Polylactic acid Polycaprolactone Polyethylene sulfonate	-	Nile blue agar plate assay; Turbidimetric assay; Molecular weight decrease by GPC.	18–40% molecular weight decrease Clear zone of PHA hydrolysis 769.23 U mg^− 1^ in LIP1 (pNPO) 714.28 U mg^−1^ in LIP1 (pNPO) 360.12 U mg^−1^ in LIP1 (PHO) 301.72 U mg^−1^ in LIP2 (PHO)	[52]
*Pseudomonas alcaligenes* LB19	Extracellular PHA depolymerase	pNP-alkanoate Poly(3-hydroxydecanoate)	Poly(3-hydroxybutyrate) Polycaprolactone Poly(L-lactide)	Monomer composition of hydrolysis products by GC chromatography.	2000 U mg^−1^ (PHO)	[59,60]
*Pseudomonas luteola* M13-4	Extracellular PHA depolymerase	pNP-alkanoate Poly(3-hydroxyalkanoate) Poly (3-hydroxybutyrate-co-3-hydroxyvalerate Poly(3-hydroxyoctanoate) Poly(3-hydroxyheptanoate) Poly(3-hydroxydodecanoate)	Poly(3-hydroxybutyrate) Polycaprolactone	Turbidimetric assay.	2.21 U mg^−1^ (Poly(3-hydroxydodecanoate)	[61]
*Bdellovibriobacteriovorus* HD100	Extracellular PHA depolymerase	Poly-(hydroxyoctanoate-cohydroxyhexanoate)	Poly(3-hydroxybutyrate)	HPLC-MS analysis of degradation products.	55 ± 2 U mg^−1^ (PHA)	[22]
*Acidovorax* sp. DP5	Extracellular PHA depolymerase	Poly-(3-hydroxybutyrate-co-4-hydroxybutyrate) [P(3HB-co-70%4HB)]	-	Weight loss.	8% weight loss 0.0075 U ml^−1^ (PHA)	[62]
*Pseudomonas mendocina* DS04-T	Extracellular Polyhydroxyalkanoate depolymerases, PHAase I, and PHAase II	Polyhydroxybutyrate Poly (3-hydroxybutyrate-co-3-hydroxyvalerate Poly-(3-hydroxybutyrate-co-4-hydroxybutyrate) [P(3HB-co-4HB)]	Poly(3-hydroxybutyrate) Polylactic acid by PHAse I	Weight loss.	1899 U mg^−1^ (PHA) (PHAase I) 893 U mg^−1^ (PHA) (PHAase II)	[63]
*Pseudomonas putida*	Intracellular PHA depolymerase, PhaZ	Poly (3-hydroxybutyrate-co-3-hydroxyvalerate Poly(3-hydroxyhexanoate) Poly(3-hydroxyoctanoate) Poly(3-hydroxynonanoate) Poly(3-hydroxydecanoate) Polylactic acid Polycaprolactone Polyethylene sulfonate	-	Nile blue agar plate assay; Turbidimetric assay; Molecular weight decrease by GPC.	Molecular weight decrease Clear zone of PHA hydrolysis 172.8 U mg^−1^ (PHO)	[17]
*Burkoldariacepacia* DP1	Extracellular PHA depolymerase	Poly-(3-hydroxybutyrate-co-4-hydroxybutyrate) [P(3HB-co-21%4HB)]	-	Weight loss.	-	[64]
*Pseudomonas chlororaphis* PA23	Intracellular lipase, LIP3 Extracellular lipase, LIP4	Poly (3-hydroxybutyrate-co-3-hydroxyvalerate Poly(3-hydroxyhexanoate) Poly(3-hydroxyoctanoate) Poly(3-hydroxynonanoate) Poly(3-hydroxydecanoate) Polylactic acid Polycaprolactone Polyethylene sulfonate	-	Nile blue agar plate assay; Turbidimetric assay; Molecular weight decrease by GPC.	49.3–49.7% molecular weight decrease Clear zone of PHA hydrolysis 625.0 U mg^−1^ in LIP3 (pNPO) 555.55 U mg^−1^ in LIP4 (pNPO) 122.13 U mg^−1^ in LIP3 (PHO) 102.6 U mg^−1^ in LIP4 (PHO)	This study
*Pseudomonas chlororaphis* PA23	Intracellular PHA depolymerase, PhaZ	Poly (3-hydroxybutyrate-co-3-hydroxyvalerate Poly(3-hydroxyhexanoate) Poly(3-hydroxyoctanoate) Poly(3-hydroxynonanoate) Poly(3-hydroxydecanoate) Polylactic acid Polycaprolactone Polyethylene sulfonate	-	Nile blue agar plate assay; Turbidimetric assay; Molecular weight decrease by GPC.	27.6% molecular weight decrease Clear zone of PHA hydrolysis 909.09 U mg^−1^ (pNPO) 650.13 U mg^−1^ (PHO)	This study

* pNP, paranitrophenyl; pNPP, paranitrophenyl palmitate; pNPO, paranitrophenyl octanoate; PHA, polyhydroxyalkanoate; PHBV, poly(3-hydroxybutyrate-co-3-hydroxyvalerate; PHO, poly(3-hydroxyoctanoate); GPC, gel permeation chromatography; FTIR, Fourier transform infrared; NMR, nuclear magnetic resonance; DSC, differential scanning calorimetry.

**Table 8 ijms-24-04501-t008:** Composition of PHA polymers used in this study.

PHA *	Substrate	* Monomer Composition of PHA (mol%)								
		3HB	3HV	3HHx	3HHp	3HO	3HN	3HD	3HDD	3HTD
PHBV	Glucose/ Valerate	76.9	23.1	ND	ND	ND	ND	ND	ND	ND
PHHx	Hexanoic acid	ND	ND	82.4	ND	16.0	ND	1.6	ND	ND
PHO	Octanoic acid	ND	ND	6.5	ND	92.0	ND	1.5	ND	ND
PHN	Nonanoic acid	ND	ND	ND	18.7	ND	81.3	ND	ND	ND
PHD	Decanoic acid	ND	ND	5.2	ND	57.4	ND	37.0	0.4	ND

* PHBV polymer was synthesized by *C. nector* H16 and all the other PHA polymers were synthesized by *P. putida* LS46 [23,25]. PHBV, poly(3-hydroxybutyrate-co-3-hydroxyvalerate; PHHx, poly(3-hydroxyhexanoate); PHO, poly(3-hydroxyoctanoate); PHN, poly(3-hydroxynonanoate); PHD, poly(3-hydroxydecanoate); ND, not detected.

**Table 9 ijms-24-04501-t009:** Primers for the construction of recombinant *lip3*, *lip4*, and *phaZ* in the pET28a vector.

Gene	Primers Used
*lip3*	FP: 5′-ATAGCTAGCTCCCAAGAGCTTGCCACGCGT-3′
	RP: 5′-ATAAAGCTTTACCCCCGCCGCTTTCAATCG-3′
*lip4*	FP: 5′-ATAGCTAGCGGGGTCGAACACAACACCCAG-3′
	RP: 5′-CCAAGCTTCTTCAGGTGTTGCTTGAGCTCT-3′
*phaZ*	FP: 5′-ATAGCTAGCCCTCAACCGTTCATCTTTCGC-3′
	RP: 5′-CCAAGCTTCGTACCGCTGAACGGTGTCGGA-3′

## Data Availability

All data related to this study are presented in the figures, tables, and Appendix A provided.

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
