# Peer review of "Polymer-Degrading Enzymes of Pseudomonas chloroaphis PA23 Display Broad Substrate Preferences"

_ijms, 2023, doi:10.3390/ijms24054501_

Round 1

Reviewer 1 Report

The article ijms-2170674 presents the characterization of thee hydrolases for degrading polymers. It was verified an expressive quantity of experiments carried out and in triplicate. The work has merits to be published in International Journal of Molecular Sciences, but some improvements are necessary.

- Abstract: ok;

- Lines 30-31: It is uncommon to insert acronyms in the keywords;

- Line 37: rewrite the expression “plasticity of its metabolic pathways”, because it is strange;

- Line 88: change to “In a previous report from our research group”;

- Line 96: PLA means “Polylactic acid” or “poly(l-lactide)”?

- Line 203-204: The expressions “alkaline pH” and “acidic pH” are inadequate. The reaction medium is alkaline or acidic, and not the pH;

- By evaluating the effect of temperature in Figure 6B, the authors must calculate the activation energy for each enzyme. By using data from Figure 6B it is possible to calculate also deactivation energy;

- Line 268: It is necessary to correct the baseline of the chromatograms, since this can harm the interpretation of the results. Furthermore, it is necessary to extend the time of the chromatograms, as there are peaks omitted after 11 min of analysis;

- The data from tables 4-6 must be more discussed. It is necessary to compare the polymer substrates;

- Conclusions: ok.

Author Response

International Journal of Molecular Science (IJMS): Manuscript ID, ijms-2170674

Response to Reviewer #1 Comments

The article ijms-2170674 presents the characterization of three hydrolases for degrading polymers. It was verified an expressive quantity of experiments carried out and in triplicate. The work has merits to be published in International Journal of Molecular

Sciences, but some improvements are necessary.

Comment 1) Abstract: ok;

Comment 2) Lines 30-31: It is uncommon to insert acronyms in the keywords

Response:  The acronyms have been removed from the keywords.

Comment 3) Line 37: Rewrite the expression “plasticity of its metabolic pathways”, because it is strange

Response: The line has been modified.

Comment 4) Line 88: Change to “In a previous report from our research group”;

Response: The sentence has been changed as suggested.

Comment 5) Line 96: PLA means “Polylactic acid” or “poly(l-lactide)”?

Response: We apologize for the mistake. PLA here refers to polylactic acid.

Comment 6) Line 203-204: The expressions “alkaline pH” and “acidic pH” are inadequate. The reaction medium is alkaline or acidic, and not the pH;

Response: The corrections have been made in the manuscript.

Comment 7) By evaluating the effect of temperature in Figure 6B, the authors must calculate the activation energy for each enzyme.

Response: The activation energy for the enzymes have been calculated and detailed in the manuscript as suggested.

Comment 8) Line 268: It is necessary to correct the baseline of the chromatograms, since this can harm the interpretation of the results. Furthermore, it is necessary to extend the time of the chromatograms, as there are peaks omitted after 11 min of analysis.

Response: The baseline was corrected at the start of each GPC run. The shift in baseline observed during the GPC run is because of the polymers (chloroform used as eluent/solvent) eluting the column. The GPC run was carried out till 15 min of analysis. The figures were constructed from 0 to 11 min as no peak was observed after 11 min for all the GPC runs.

Comment 9) The data from Tables 4-6 must be more discussed. It is necessary to compare the polymer substrates

Response: The data from tables 4-6 has been discussed in the manuscript as suggested.

Comment 10) Conclusions: ok.

Reviewer 2 Report

The paper titled “Characterization of polymer-degrading lipases and a polyhydroxyalkanoate depolymerase from Pseudomonas chlororaphis PA23” characterized the degrading lipases of chlororaphis. This work has done a classical work process and can be published in IJMS.

I have only small comments on this manuscript.

1. why the enzymatic activity shows a peak during the pH variation?  Is this due to the decomposition of LIP3 and LIP4 at low/high pH? Or they are not compatible same with the pH? In figure 6

2. it seems the enzymatic activity of the three is nearly the same with the temperature. Why? In figure 6

3. please mark the difference of B and C in figure 6.

4. The reactivity in figure 7 and 6 are 100%. Is this number so high?

5. some related papers need to be cited “Supervariate ceramics: biomineralization mechanism." Materials Today Advances 10 (2021): 100144.”

Author Response

International Journal of Molecular Science (IJMS): Manuscript ID, ijms-2170674

Response to Reviewer Reviewer #2 Comments

The paper titled “Characterization of polymer-degrading lipases and a polyhydroxyalkanoate depolymerase from Pseudomonas chlororaphis PA23” characterized the degrading lipases of chlororaphis. This work has done a classical work process and can be published in IJMS.

Comment 1) Why does the enzymatic activity show a peak during the pH variation? Is this due to the decomposition of LIP3 and LIP4 at low/high pH? Or they are not compatible same with the pH? In Figure 6.

Response: The enzymes exhibit different activity with varying pH as the pH difference causes structural changes and therefore leads to differences in the binding of substrates to the active site residues of the enzymes, thus affecting enzyme activity. As depicted in the figure, the enzymes exhibited different activity with varying pH conditions because of the differences in the arrangement of active site residues involved in binding and hydrolysis of substrate.

Comment 2) It seems the enzymatic activity of the three is nearly the same with the temperature. Why? In Figure 6.

Response: The enzymes exhibited optimum activity at moderate temperature with an optimum temperature of 45 – 50 °C. The most probable reason of these enzyme exhibiting similar temperature optima is that the microbial source of these enzymes is a mesophilic organism, Pseudomonas chlororaphis, which grows and thrives best in a moderate temperature range of 25 – 45 °C. Though, most of the enzymes produced by mesophilic organisms produce enzymes with optimum activity at moderate temperatures, there are some exceptions which can be explained because of the difference in the structural folding and conformation of the enzyme with respect to the change in temperature.

Comment 3) Please mark the difference of B and C in Figure 6.

Response: The difference of Figure 6B and 6C has been highlighted in figure legend

Comment 4) The reactivity in Figure 7 and 6 are 100%. Is this number so high?

Response: Figures 6 and 7 show the relative activity of the enzymes corresponding to the observed values obtained for pH, temperature, and substrate specificity. A value of 100% represents the maximum activity (not the actual value) relative to other values obtained.

Comment 5) Some related papers need to be cited “Supervariate ceramics: biomineralization mechanism." Materials Today Advances 10 (2021): 100144.”

Response: The article plus two additional related references, have been cited in the manuscript, as suggested.

Reviewer 3 Report

Reviewer’s Comments:

The manuscript “Characterization of polymer-degrading lipases and a polyhydroxyalkanoate depolymerase from Pseudomonas chlororaphis PA23” is a very interesting work. In this work, although many bacterial lipases and PHA depolymerases have been identified, cloned, and characterized, there is very little information on the potential application of lipases and PHA depolymerases, especially intracellular enzymes, for the degradation of polyester polymers/plastics. We identified genes encoding an intracellular lipase (LIP3), an extracellular lipase (LIP4), and an intracellular PHA depolymerase (PhaZ) in the genome of the bacterium, Pseudomonas chlororaphis PA23. We cloned these genes into Escherichia coli and then expressed, purified, and characterized the biochemistry and substrate preferences of the enzymes they encode. Our data suggest that the LIP3, LIP4, and PhaZ enzymes differ significantly in their biochemical and biophysical properties, structural-folding characteristics, and the absence or presence of a lid domain. Despite their different properties, the enzymes exhibited broad substrate specificity and were able to hydrolyze both short- and medium-chain length polyhydroxyalkanoates (PHAs), para-nitrophenyl (pNP) alkanoates, and polylactic acid (PLA). While I believe this topic is of great interest to our readers, I think it needs major revision before it is ready for publication. So, I recommend this manuscript for publication with major revisions.

1. In this manuscript, the authors did not explain the importance of the Pseudomonas chlororaphis in the introduction part. The authors should explain the importance of Pseudomonas chlororaphis.

2) Title: The title of the manuscript is not impressive. It should be modified or rewritten it.

3) Correct the following statement “Gel Permeation Chromatography (GPC) analyses showed a decrease in the molecular weight of the polymers after incubation with LIP3, LIP4, and PhaZ. The enzymes also showed some polymer-degrading activity on petroleum-based polyester polymers such as poly(ε-caprolactone) (PCL) and polyethylene succinate (PES), suggesting that these enzymes may be useful for biodegradation of synthetic polyester plastics”.

4) Keywords: The Pseudomonas chlororaphis is missing in the keywords. So, modify the keywords.

5) Introduction part is not impressive. The references cited are very old. So, Improve it with some latest literature like 10.1016/j.molstruc.2021.131145, 10.3389/fchem.2022.1023316

6) The authors should explain the following statement with recent references, “The GPC data also support the observations that the LIP3, LIP4, and PhaZ enzymes 280
were able to hydrolyze ester bonds of a wide variety of polymer types”.

7) Add space between magnitude and unit. For example, in synthesis “21.96g” should be 21.96 g. Make the corrections throughout the manuscript regarding values and units.

8) The author should provide reason about this statement “Lipases and PHA depolymerases are generally highly variable in size, and sequence similarity between them is limited to short regions located around active-site residues”.

9. Comparison of the present results with other similar findings in the literature should be discussed in more detail. This is necessary in order to place this work together with other work in the field and to give more credibility to the present results.

10) Conclusion part is very long. Make it brief and improve by adding the results of your studies.

11) There are many grammatic mistakes. Improve the English grammar of the manuscript.

Author Response

International Journal of Molecular Science (IJMS): Manuscript ID, ijms-2170674

Response to Reviewer Reviewer #3 Comments

The manuscript “Characterization of polymer-degrading lipases and a polyhydroxyalkanoate depolymerase from Pseudomonas chlororaphis PA23” is a very interesting work. In this work, although many bacterial lipases and PHA depolymerases have been identified, cloned, and characterized, there is very little information on the potential application of lipases and PHA depolymerases, especially intracellular enzymes, for the degradation of polyester polymers/plastics. We identified genes encoding an intracellular lipase (LIP3), an extracellular lipase (LIP4), and an intracellular PHA depolymerase (PhaZ) in the genome of the bacterium, Pseudomonas chlororaphis PA23. We cloned these genes into Escherichia coli and then expressed, purified, and characterized the biochemistry and substrate preferences of the enzymes they encode. Our data suggest that the LIP3, LIP4, and PhaZ enzymes differ significantly in their biochemical and biophysical properties, structural-folding characteristics, and the absence or presence of a lid domain. Despite their different properties, the enzymes exhibited broad substrate specificity and were able to hydrolyze both short- and medium-chain length polyhydroxyalkanoates (PHAs), paranitrophenyl (pNP) alkanoates, and polylactic acid (PLA). While I believe this topic is of great interest to our readers, I think it needs major revision before it is ready for publication. So, I recommend this manuscript for publication with major revisions.

Comment 1) In this manuscript, the authors did not explain the importance of the Pseudomonas chlororaphis in the introduction part. The authors should explain the importance of Pseudomonas chlororaphis.

Response:   The importance of P. chlororaphis has been added in the manuscript as suggested.

Comment 2) Title: The title of the manuscript is not impressive. It should be modified or rewritten it.

Response:  Suggested title: “Polymer-degrading enzymes of Pseudomonas chloroaphis PA23 display broad substrate preferences”

Comment 3) Correct the following statement “Gel Permeation Chromatography (GPC) analyses showed a decrease in the molecular weight of the polymers after incubation with LIP3, LIP4, and PhaZ. The enzymes also showed some polymer-degrading activity on petroleum-based polyester polymers such as poly(ε- caprolactone) (PCL) and polyethylene succinate (PES), suggesting that these enzymes may be useful for biodegradation of synthetic polyester plastics”.

Response: The statement has been corrected as suggested.

Comment 4) Keywords: The Pseudomonas chlororaphis is missing in the keywords. So, modify the keywords.

Response: Pseudomonas chlororaphis has been added in the manuscript.

Comment 5) Introduction part is not impressive. The references cited are very old. So, improve it with some latest literature like 10.1016/j.molstruc.2021.131145, 10.3389/fchem.2022.1023316

Response:  The Introduction has been revised and more recent references have been cited.

Comment 6) The authors should explain the following statement with recent references, “The GPC data also support the observations that the LIP3, LIP4, and PhaZ enzymes were able to hydrolyze ester bonds of a wide variety of polymer types”.

Response: The recent references have been added in the manuscript and a new Table (Table 7) has been added detailing the bacterial lipases and PHA depolymerases capable of degrading different polymer types.

Comment 7) Add space between magnitude and unit. For example, in synthesis “21.96g” should be 21.96 g. Make the corrections throughout the manuscript regarding values and units.

Response: The corrections have been made in the manuscript as suggested.

Comment 8) The author should provide reason about this statement “Lipases and PHA depolymerases are generally highly variable in size, and sequence similarity between them is limited to short regions located around active-site residues”.

Response: The lipases and PHA depolymerases are known to have different molecular weights within each enzyme class and are mostly similar around the active site regions as shown in Figures 3, 4 and 5. The literatures supporting the statement have also been added in the manuscript.

Comment 9) Comparison of the present results with other similar findings in the literature should be discussed in more detail. This is necessary in order to place this work together with other work in the field and to give more credibility to the present results.

Response: A table (Table 9) has been added in the manuscript showing the lipases and PHA depolymerises from bacterial species which are reported to degrade polyesters. As shown in the Table, LIP3, LIP4, and PhaZ are much more effective in degrading a large number of polymers including pNP-alkanoates, PHAs, and petrochemical derived polymers such as PES and PCL (broad substrate spectrum), as compared to others. Further, majority of the studies have been done using extracellular lipases and PHA depolymerases; there is only few reports on the complete characterization of intracellular lipases and PHA depolymerises from bacterial source for biodegradation of polymers.

Comment 10) Conclusion part is very long. Make it brief and improve by adding the results of your studies.

Response: The Conclusion has been improved, as suggested

Comment 11) There are many grammatic mistakes. Improve the English grammar of the manuscript.

Response: The revised manuscript has been carefully proof-read and the grammatical errors have been corrected.

Round 2

Reviewer 1 Report

Dear authors,

The article has been improvised, but further corrections are needed, which are commented in the attached file.

Author Response

Comment 1) Change in all text: "M" to "molL-1".

Response: The "M" has been changed to to "molL-1" in the manuscript.

Comment 2) The more appropriate substrate to determining lipase activity is a vegetable oil, which is rich in triglycerides. This aspect must be discussed in the text.

Response: Reference to the hydrolysis of triglycerides by lipases has been included in the Introduction, on page 3, lines 34 to 37 in the “marked-up” version.

Comment 3) It is necessary to mention in the text that no peak was observed after 11 min of elution for all GPC analyses.

Response: The following sentence has been inserted. “No peak was observed after 11 minutes of elution for all GPC runs”, on Page 18, lines 314-315.
